# ViT-V-Net: Vision Transformer for Unsupervised Volumetric Medical Image Registration

**Junyu Chen**[1,2]                                                                          JCHEN245@JHMI.EDU

**Yufan He**[1]                                                                                  YHE35@@JHU.EDU

**Eric C. Frey**[1,2]                                                                          EFREY@JHMI.EDU

**Ye Li**[1,2]                                                                                    YLI192@@JHU.EDU

**Yong Du**[2]                                                                                  DUYONG@JHU.EDU

[1] *Department of Electrical and Computer Engineering, Johns Hopkins University, USA*

[2] *Department of Radiology and Radiological Science, Johns Hopkins Medical Institutes, USA*

## Abstract

In the last decade, convolutional neural networks (ConvNets) have dominated and achieved state-of-the-art performances in a variety of medical imaging applications. However, the performances of ConvNets are still limited by lacking the understanding of long-range spatial relations in an image. The recently proposed Vision Transformer (ViT) for image classification uses a purely self-attention-based model that learns long-range spatial relations to focus on the relevant parts of an image. Nevertheless, ViT emphasizes the low-resolution features because of the consecutive downsamplings, result in a lack of detailed localization information, making it unsuitable for image registration. Recently, several ViT-based image segmentation methods have been combined with ConvNets to improve the recovery of detailed localization information. Inspired by them, we present ViT-V-Net, which bridges ViT and ConvNet to provide volumetric medical image registration. The experimental results presented here demonstrate that the proposed architecture achieves superior performance to several top-performing registration methods.

**Keywords:** Image Registration, Vision Transformer, Convolutional Neural Networks.

## 1. Introduction

Deformable image registration (DIR) is fundamental for many medical image analysis tasks. It functions by of establishing spatial correspondences between points in a pair of fixed and moving images through a spatially varying deformation model. Traditionally, DIR can be performed by solving an optimization problem that maximizes the image similarity between the deformed moving and fixed images while enforcing smoothness constraints on the deformation field (Avants et al., 2008; Modat et al., 2010). However, such optimization problems need to be solved for each pair of images, making those methods computationally expensive and slow in practice. Since recently, ConvNets-based methods (Balakrishnan et al., 2018) have become a major focus of attention due to their fast computation time after training while achieving comparable accuracy to state-of-the-art methods.

Despite ConvNets' promising performance, ConvNet architectures generally have limitations in modeling explicit long-range spatial relations (i.e., relations between two voxels that are far away from each other) present in an image due to the intrinsic locality of convolution operations (Chen et al., 2021). Many works have been proposed to overcoming this limitation, e.g. V-Net (Milletari et al., 2016) and self-attention (Dosovitskiy et al., 2020). Recently, there has been an increasing interest in developing self-attention-based architectures due to their great success in natural language processing. Dosovitskiy et al. (Dosovitskiy et al., 2020) proposed Vision Transformer (ViT), a first purely self-attention-based network, and achieved state-of-the-art performance in image recognition. Subsequent

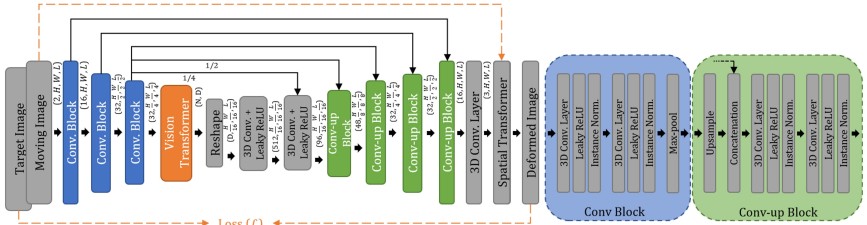

**Figure 1:** Method overview and network architecture of ViT-V-Net.

to this progress, TransUnet (Chen et al., 2021) was developed on the basis of a *pre-trained* ViT for 2-dimensional (2D) medical image segmentation. In this work, we present the first study to investigate the usage of ViT for volumetric medical image registration. We propose ViT-V-Net that employs a hybrid ConvNet-Transformer architecture for self-supervised volumetric image registration. In this method, the ViT was applied to high-level features of moving and fixed images, which required the network to learn long-distance relationships between points in images.

## 2. Methods

Let $f \in \mathbb{R}^{H \times W \times L}$ and $m \in \mathbb{R}^{H \times W \times L}$ be fixed and moving image volumes. We assume that $f$ and $m$ are single-channel grayscale images, and they are affinely aligned. Our goal is to predict a transformation function $\phi$ that warps $m$ (i.e., $m \circ \phi$) to $f$, where $\phi = Id + \mathbf{u}$, $\mathbf{u}$ denotes a flow field of displacement vectors, and $Id$ denotes the identity. Fig. 1 presents an overview of our method. First, the deep neural network ($g_\theta$) generates $\mathbf{u}$, for the given image pair $f$ and $m$, using a set of parameters $\theta$ (i.e., $\mathbf{u} = g_\theta(f, m)$). Then, the warping (i.e., $m \circ \phi$) is performed via a spatial transformation function. During network training, image similarity between $m \circ \phi$ and $f$ is compared, and the loss is backpropagated into the network.

**ViT-V-Net Architecture** Naive application of ViT to full-resolution volumetric images leads to large computational complexity. Here, instead of feeding full-resolution images directly into the ViT, the images (i.e., $f$ and $m$) were first encoded into high-level feature representations via a series of convolutional layers and max-poolings. In the ViT, the high-level features were then separated into $N$ vectorized $P^3 \times C$ patches, where $N = \frac{HWL}{P^3}$, $P$ denotes the patch size, and $C$ is the channel size. Next, the patches were mapped to a latent $D$-dimensional space using a trainable linear projection (i.e., patch embedding). Learnable position embeddings are then added to the patch embeddings to retain positional information of the patches (Dosovitskiy et al., 2020). Next, the resulting patches were fed into the Transformer encoder, which consisted of 12 alternating layers of Multihead Self-Attention (MSA) and Multi-Layer Perceptron (MLP) blocks (see (Dosovitskiy et al., 2020; Chen et al., 2021) for details of ViT). Finally, the output from ViT was reshaped and then decoded using a V-Net style decoder. Notice that long skip connections between the encoder and decoder were also used. The network's final output is a dense displacement field, which was then used in the spatial transformer for warping $m$.

**Loss Functions** The image similarity measurement used in this study was mean squared error (MSE), along with a diffusion regularizer controlled by a weighting parameter $\lambda$ for imposing smoothness in the displacement field $\mathbf{u}$.

|  | Affine only | NiftyReg | SyN | VoxelMoprh-1 | VoxelMorph-2 | ViT-V-Net |
|---|---|---|---|---|---|---|
| **Dice** | 0.569±0.171 | 0.713±0.134 | 0.688±0.140 | 0.707±0.137 | 0.711±0.135 | **0.726±0.130** |

**Table 1:** Overall Dice comparisons between the proposed method and the others.

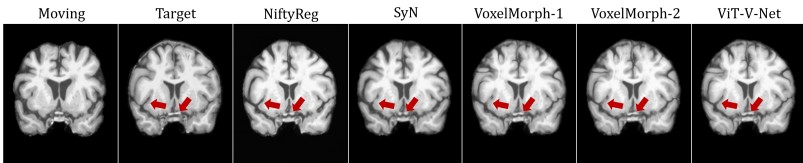

**Figure 2:** Registration results of a MR coronal slice.

## 3. Results and Conclusions

We demonstrate our method on the task of brain MRI registration. We used an in-house dataset that consists of 260 T1–weighted brain MRI scans. The dataset was split into 182, 26, and 52 (7:1:2) volumes for training, validation, and test sets. Each image volume was randomly matched to two other volumes to form four pairs of $f$ and $m$, resulting in 768, 104, and 208 image pairs. Label maps including 29 anatomical structures were obtained using FreeSurfer (Fischl, 2012) for evaluation. The proposed method was compared in terms of Dice score to Symmetric Normalization (SyN), NiftyReg, and VoxelMorph-1 and -2 (Balakrishnan et al., 2018). The regularization parameter, $\lambda$, was set to be 0.02, which was reported in VoxelMorph as an optimal value. Qualitative results, and Dice scores are shown in Table 1 and Fig. 2. As visible from the results, the proposed ViT-V-Net yielded a significant gain of $> 0.1$ in Dice performance ($p$-values $< 0.005$) compared to the others. We also noticed that ViT-V-Net reached lower loss values and had higher validation Dice scores during training. In conclusion, the proposed ViT-based architecture achieved superior performance than the top-performing registration methods, demonstrating the effectiveness of ViT-V-Net.

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
