# OpenReview forum: "ViT-V-Net: Vision Transformer for Unsupervised Volumetric Medical Image Registration"
_MIDL.io/2021/Conference/Short — MIDL 2021 Poster_

### Official Review · Reviewer_zrnK · 2021-04-23

**Confidence:** 3
**Final Rating:** 3

**Summary:**

Considering the limitation of  ConvNets in lacking the understanding of long-range spatial relations, the Vision Transformer(ViT) was introduced into this paper. The authors proposed ViT-V-Net for self-supervised volumetric medical image registration. The results show the proposed method is better than other baselines.

**Strengths:**

1. From the authors' knowledge, it is probably one of the first attempts to apply transformer architecture in self-supervised volumetric image registration.

2.  The results are better compared with other baselines.



**Weaknesses:**

1. Attention module plays the same role as the transformer. The comparison experiments between them are beneficial to highlight the superiority of the transformer.

2. Since there exist some methods that could tackle the long-range spatial relations, the hidden motivation of the transformer should be more explicit and clearer.

**Deanonymize Review:**

no

**Detailed Comments:**

See in the weakness

**Justification Of The Rating:**

The overall quality is above the acceptance line. But some issues should be claimed, and some comparisons between the attention module and transformer should be added here. The rate will be adjusted based on the rebuttal process.

**Paper Type:**

validation/application paper

**Special Issue:**

no

---

### Official Review · Reviewer_LWXp · 2021-04-30

**Confidence:** 3
**Final Rating:** 2

**Summary:**

The paper presents an image registration method that uses Vision Transformers to tackle the problem of long-range spatial relations. The authors evaluated their method on an in-house MR brain dataset (T1-T1 registration). They show that their method performs slightly better than Voxelmorph (Dice 0.726 to 0.711)

**Strengths:**

-The authors firstly  used Vision Transformers in the context of image registration
- the code is available
- The method shows promising results on the in-house dataset (Dice of 0.726 compared to 0.711 of Voxelmorph)

**Weaknesses:**

- The authors used some \vskip or any other format/spacing altering commands (see the spacing between abstract and introduction or any other new section) Without those commands the paper limit of 3 pages is not met.
- The authors only evaluate the accuracy of the presented method using the Dice Score. An evaluation regarding the regularity of the deformation field (e.g. number of foldings) is missing.
- References to dl-based multi-level image registration approaches are missing - this is one way to tackle to problem of large motions.
- the method is not evaluated on a public dataset.
- no clinical relevant application
- There are some points that are not clear, e.g. what is the difference between Voxelmorph 1 and 2?
- The authors say that the results are significant with a p-value<0.005, however, they do not say which test they used.

**Deanonymize Review:**

no

**Justification Of The Rating:**

The paper presents an interesting method that is relevant and interesting for the MIDL community and therefore, the paper could be presented at MIDL 2021. However, the 3-page limit is hurt due to the format/spacing altering commands. Therefore, I weak reject the paper and the ACs/PCs have to decide how to proceed.

**Paper Type:**

methodological development

**Special Issue:**

no

---

### Meta-Review · Area_Chair_eS4R · 2021-05-10

**Recommendation:** Accept (Poster)
**Confidence:** 5

**Metareview:**

Both reviewers favorably comment on novelty. While the evaluation could be improved, the ideas are worth discussing at MIDL. I will accept, provided that the authors fix the style sheet transgressions in the final version.

---

### Decision · Program_Chairs · 2021-05-11

Accept (Poster)